# Biomass Change Estimated by TanDEM-X Interferometry and GEDI in a Tanzanian Forest

**Svein Solberg** [1,*][iD]**, Ole Martin Bollandsås** [2]**, Terje Gobakken** [2][iD]**, Erik Næsset** [2][iD]**, Paromita Basak** [3]
**and Laura Innice Duncanson** [3]

1   Norwegian Institute of Bioeconomy Research (NIBIO), NO-1431 Ås, Norway
2   Faculty of Environmental Sciences and Natural Resource Management, Norwegian University of Life Sciences (NMBU), NO-1432 Ås, Norway; ole.martin.bollandsas@nmbu.no (O.M.B.); terje.gobakken@nmbu.no (T.G.); erik.naesset@nmbu.no (E.N.)
3   Department of Geographical Sciences, University of Maryland College Park, College Park, MD 20742, USA; pbasak@umd.edu (P.B.); lduncans@umd.edu (L.I.D.)
*   Correspondence: svein.solberg@nibio.no; Tel.: +47-92853902

**Abstract:** Mapping and quantification of forest biomass change are key for forest management and for forests' contribution to the global carbon budget. We explored the potential of covering this with repeated acquisitions with TanDEM-X. We used an eight-year period in a Tanzanian miombo woodland as a test case, having repeated TanDEM-X elevation data for this period and repeated field inventory data. We also investigated the use of GEDI space–LiDAR footprint AGB estimates as an alternative to field inventory. The map of TanDEM-X elevation change appeared to be an accurate representation of the geography of forest biomass change. The relationship between TanDEM-X phase height and above-ground biomass (AGB) could be represented as a straight line passing through the origin, and this relationship was the same at both the beginning and end of the period. We obtained a similar relationship when we replaced field plot data with the GEDI data. In conclusion, temporal change in miombo woodland biomass is closely related to change in InSAR elevation, and this enabled both an accurate mapping and quantification wall to wall within 5–10% error margins. The combination of TanDEM-X and GEDI may have a near-global potential for estimation of temporal change in forest biomass.

**Keywords:** InSAR; DEM; temporal change; forest; biomass

## 1. Introduction

Forest generates a suite of ecosystem services, including wood production and carbon storage. Monitoring growing stock and above-ground biomass (AGB) is valuable for a range of purposes such as forest management and global carbon accounting and carbon cycle modeling [1]. Deforestation and forest degradation in the tropics contribute considerable fractions of anthropogenic greenhouse gas emissions [2], and performance-based payment is one way to reduce this problem and requires forest monitoring data at large scale [3].

Satellite remote sensing can provide data at large scale for forest monitoring. Deforestation areas can be mapped with optical and SAR sensors, and associated forest carbon losses can be estimated by combining this with carbon density data for the given type of forest [4–6]. These methods can have limitations and discrepancies [7,8]. Another approach is to use satellite LiDAR full-waveform footprints of Global Ecosystem Dynamics Investigation (GEDI) for sampling-based AGB estimation [9].

In this study we employ X-band interferometric SAR (InSAR). TanDEM-X is a satellite mission that provides single-pass, across-track interferometry, offering sufficiently high coherence to enable extraction of accurate phase height in forest. With the short wavelength, we obtain a phase center high up in the forest canopy and thereby a sensitive method for mapping temporal change in forest canopy height and associated AGB change. The idea is

that phase height decrease and increase are effects of disturbance and growth, respectively, and that the corresponding change in AGB can be estimated with a conversion factor from phase height to AGB.

The capability of this approach has not been fully demonstrated yet; however, it is timely now because a second, global elevation model based on TanDEM-X is underway which, in combination with the former global DEM from 2012, will provide phase height change, and possibly estimates of forest AGB change, at high resolution globally. The approach is based on several studies. First, X-band InSAR phase height above ground is related to forest AGB and other forest attribute data in a reasonably linear way [10–15], and it has been shown that forest AGB in some cases can be mapped equally as accurately as with airborne laser scanning [16]. Secondly, studies of change between SRTM and TanDEM-X elevation models, or between repeated TanDEM-X data, have shown that the phase height change provides change maps that correspond well with known logging and thinning areas [15,17], with the spatial distribution of the loss category in global data sets based on Landsat [4] and with the changes seen in protected and unprotected forest areas [18,19] and, finally, with change maps based on repeated airborne laser scanning [20]. However, errors in the SRTM data and differences in wavelength and acquisition geometries have resulted in uncertainty, low accuracy and bias in estimates of AGB change based on SRTM and TanDEM-X elevation models [18,20]. Finally, while this approach relies on a stable relationship between InSAR phase height and AGB, this is not always the case [21]. The relationship changes considerably between frozen and unfrozen conditions because the X-band microwaves penetrate deeper down into a frozen forest canopy than an unfrozen one [22]. We assume also that the penetration rate differs between leaf-on and leaf-off canopy conditions.

In this study, we intend to explore the full potential of this approach and have novel elements in place for that. First, we now have TanDEM-X data repeated at two points in time. Secondly, we have repeated field inventory data with AGB estimates at the same points in time. The study area is a Tanzanian forest, and such repeated field data are possibly unique for a tropical forest and valuable both for estimating the conversion factor between phase height and AGB and for checking that the relationship is stable and comparable at both points in time. Thirdly, we explore the use of GEDI as an alternative to field inventory in combination with TanDEM-X in a novel way. There is ongoing research on the fusion of TanDEM-X and GEDI for estimation of forest height, structure and biomass [23,24]; however, our approach in the present study is complementary to that. The present approach has prerequisites, which are that the relationship between phase height and AGB is straight and linear, that it passes through origin and that it is similar at both points in time. The idea is to derive the conversion factor from a relationship by combining data from TanDEM-X with the field inventory plots or the GEDI footprints, both having known terrain heights.

$$AGB = k \, IH_{TDX} \tag{1}$$

where $k$ is the conversion factor, $IH$ is the interferometric phase height above ground and the subscript $TDX$ refers to TanDEM-X. This is used to estimate temporal change of AGB based on temporal change of elevation H as follows:

$$\Delta AGB = k \Delta H_{TDX} \tag{2}$$

In addition, we employ a new method for estimating the conversion factor from phase height to AGB.

The primary objective of this study was to investigate forest biomass change derived from repeated TanDEM-X elevation data sets, including the accuracy of a change map and a wall-to-wall estimate. A secondary objective was to investigate whether GEDI footprint AGB estimates could replace field inventory plots as ground truth and calibration data.

## 2. Materials and Methods

We selected a rectangular study area in Liwale in the Lindi region, Tanzania (Figure 1). The study area was 45 km × 45 km. The area has woodlands of the miombo type with high trees and shrubs and grasses on the forest floor. It varies between wet and dry vegetation types, and the dominating tree species are *Brachystegia* sp., *Julbernadia* sp. and *Pterocarpus angolensis*. The climate typically has four seasons, i.e., a wet period in November–January; a dry February; a wet period in March–May; and, finally, a dry period in June–October. Mean annual temperature range is 20–30 °C, and annual rainfall range is 600–1000 mm. We used the spatial reference UTM36S for processing and visualization of the data sets.

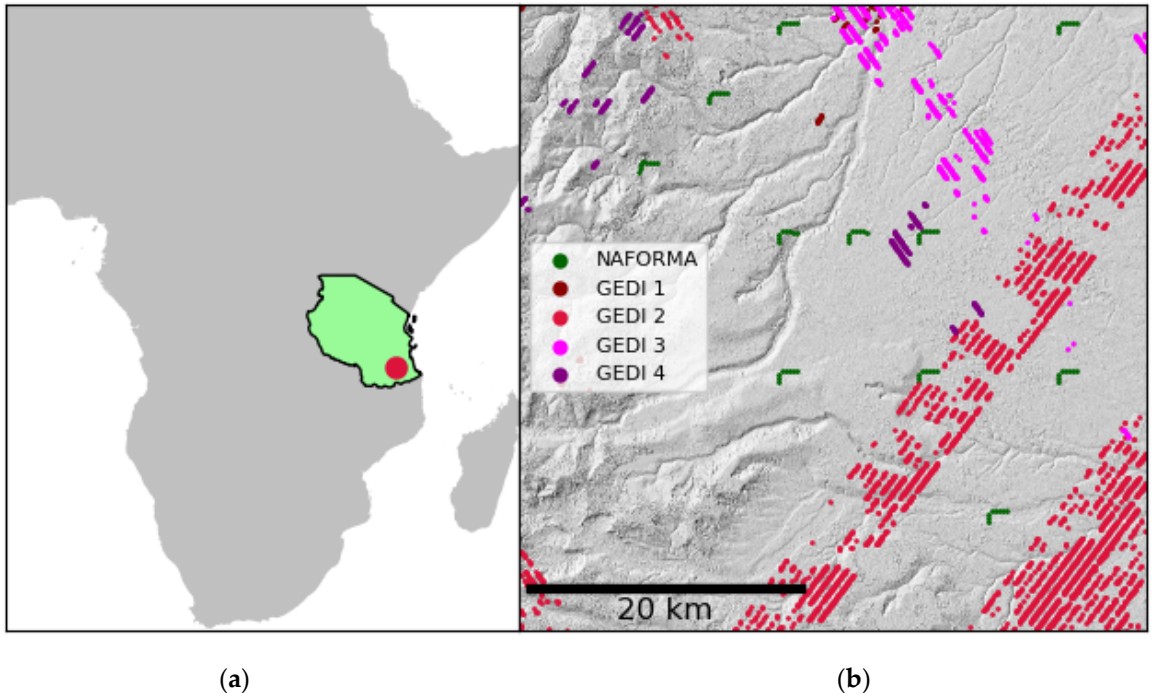

(**a**)　　　　　　　　　　　　　　　　　　　　　　(**b**)

**Figure 1.** (**a**) The location of the study area in Tanzania, Africa, and (**b**) the 45 km × 45 km area with NAFORMA field plot clusters and GEDI footprints shown on a background of a hill-shade grayscale elevation model of the TanDEM-X global DEM from 2012. The time periods for GEDI in 2019 and 2020 were as follows: GEDI 1 = 19 December–20 January; GEDI 2 = April–20 May; GEDI 3 = 19 January and 20 July; and GEDI 4 = 19 July.

### 2.1. NAFORMA National Forest Inventory Data

Our first and main reference data set was a subset of the National Forestry Resource Monitoring and Assessment Program of Tanzania (NAFORMA) plots. We used this data set both to estimate a conversion factor between InSAR height and AGB and to have a ground truth for AGB change. This is possibly a unique data set for the tropics as it has AGB at two points in time based on field measurements. The data set comprised 11 L-shaped clusters each with 8 plots, totaling 88 plots (Figure 1). The measurements were carried out in February–June 2012 [25] as part of the national campaign, and they were repeated in February 2020 as part of the present project. This provided a period of eight years of growth. The plot size was 707 m$^2$. Each plot contained a set of 2, 5, 10 and 15 m radius concentric circles where all trees with diameters at breast height (DBHs) exceeding 1, 5, 10 and 20 cm, respectively, were recorded [26]. A tree was measured if it was taller than or equal to 1.35 m and capable of reaching at least 5 m in height. Cactus, palm, bamboo and shrubs were not recorded. DBH was measured on each tree, and the measurements at the two points in time corresponded well, having strong correlations (R$^2$ > 0.99). Tree height was measured for every fifth tree. For trees without measured height, the height was predicted using diameter–height models. These models were fitted using height sample

trees from plots across the entire Liwale region, measured in 2012, including the plots selected for the current study. However, we limited our analysis to plots within the same NAFORMA strata as the 11 clusters within our study area. Specifically, we retained plots from NAFORMA strata 5, 6, 7 and 8 [27]. To construct the diameter–height models, we combined strata 5 and 6 (representing low-volume plots) and strata 7 and 8 (representing high-volume plots), creating separate models for each merged stratum. In total, our data set comprised 1173 height sample trees, with 398 and 775 distributed among low- and high-volume plots, respectively. The models took the form of h = 1.3 + $\alpha \times$ DBH$^\beta$, where $\alpha$ and $\beta$ are the model parameters. The goodness of fit for these models, as indicated by $R^2$ and root-mean-square error relative to the mean observed biomass (RMSE, %), were 0.54 and 36% for the low-volume plots and 0.58 and 32% for the high-volume plots. We predicted the biomass of each tree using allometric biomass models developed by Mugasha et al. [28], incorporating both field-measured DBH and predicted height as input variables. Plot-wise AGB estimates were obtained by summing the biomass predictions for each tree and scaling to per hectare values. The field data contained terrain elevation.

## 2.2. GEDI

Our second reference data set was 3890 footprints acquired with the GEDI spaceborne LiDAR sensor (Figure 1), which served as an alternative for estimating the conversion factor between InSAR height and AGB. The GEDI instrument comprises three lasers generating eight ground transects, each with ~25 m diameter footprints spaced approximately every 60 m along the track and with 600 m between transects. For AGB, we used the GEDI Level 4A (L4A) AGBD (Mg/ha) data, version 2.1 [29], which were provided for each footprint. This AGBD variable had been obtained through parametric models based on simulated GEDI Level 2A (L2A) waveform relative height (RH) metrics from airborne LiDAR and ground-based measurements of AGB [30]. There were distinct models for various regions and plant functional types [29]. The model most applicable to this study area was Deciduous Broadleaf Trees Africa [31]. For InSAR height, we used the terrain elevation variable provided with the footprint data representing the elevation of the ground underneath the canopy, which we subtracted from the TanDEM-X DEM. We selected GEDI data based on both quality and time of acquisition to overlap with the leaf-on season as close to the TanDEM-X acquisition from December 2019 as possible. This led to the identification of four prioritized time periods: (a) December–20 January 2019; (b) April–May 2020; (c) June 2020; and (d) July 2019. To ensure data quality, we used a quality filter requiring that the waveforms should have L2A and L4A quality flags = 1 and beam sensitivity and geolocation/sensitivity_a2 > 0.95 to ensure sufficient signal-to-noise ratio (SNR) to penetrate canopies with up to at least 95% canopy cover, which would be sufficient for detecting both ground under dense canopy and the top of sparse canopies.

## 2.3. TanDEM-X

TanDEM-X is a Synthetic Aperture Radar (SAR) interferometer based on two satellites moving in close formation and generating across-track, single-pass acquisitions which is mainly used to generate digital surface models (DSMs). The wavelength is X-band = 3.1 cm. This short wavelength has limited penetration ability down into the forest canopy, and it results in a DSM close to the top of the canopy. We used TanDEM-X DSMs at two points in time, coinciding fairly well with the timing of the field plot inventories. We obtained the data from German Aerospace Center's (DLR) data-sharing platform EOWEB. The TanDEM-X data used in this study were all acquired in dual-pol (HH), stripmap mode.

For the first point in time, we used the two global DEM tiles S10E037 and S10E038. We used the 0.4-arcsecond spatial resolution version. We combined the two tiles by mosaicking and resampled them to 5 m resolution and UTM36S using bilinear interpolation. The two tiles were composed of acquisitions covering a period of almost three years, i.e., between December 2010 and September 2013. Based on the EOWEB archive and the metadata of the tiles, we identified the mean date of the acquisitions for our study area to be in May 2012.

For the second point in time, no DSM was available. A TanDEM-X-based DEM ("Change-DEM") is under construction. We generated a DSM for the study area based on three suitable TanDEM-X data sets from December 2019 that covered almost the entire study area. The data were obtained as single-look complex data provided in the CoSSC format. For convenience, we have labeled them A, B and C (Table 1). They all had a height of ambiguity (HoA) of around 55 m.

**Table 1.** TanDEM-X CoSSC data sets with date of acquisition and height of ambiguity (m).

| Data Set | Date | HoA |
|----------|------|-----|
| A | 20 December 2019 | 55.7 |
| B | 31 December 2019 | 55.2 |
| C | 31 December 2019 | 55.3 |

*2.4. TanDEM-X Processing*

The processing from SLC to DSM consisted of a sequence of processing steps where we used ENVI version 5.6.1 and Sarscape version 5.6 software (Figure 2). For each acquisition, we first generated an interferogram by combining the complex data of the two satellites. We worked towards a final product with 5 m × 5 m resolution and used a multi-looking value of 2 in azimuth and 1 or 2 in range. We used the TanDEM-X global DEM from 2012 as reference, i.e., to generate a synthetic interferogram, which we subtracted from the INT, and derived a differential interferogram. This represented the height change for the period 2012–2019, which was dominated by forest height changes caused by logging and growth. Some terrain elevation changes due to construction might have happened; however, we have considered this as of minor influence here. We filtered the differential interferogram with a Goldstein filter [32] to reduce phase noise. We carried out a phase unwrapping of this using the Minimum Cost Flow method and geocoded this to 5 m × 5 m resolution using bilinear interpolation.

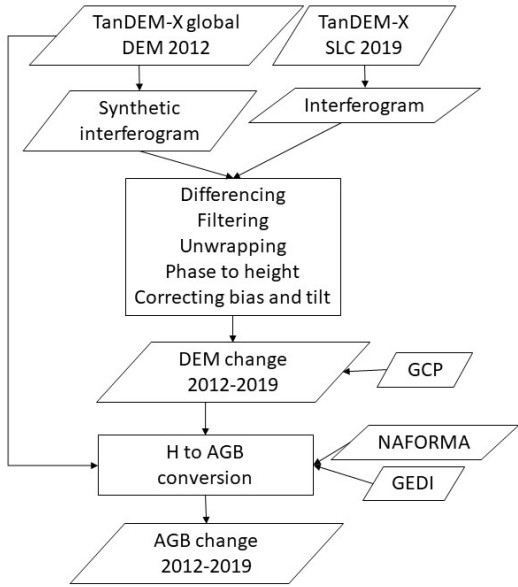

**Figure 2.** Flowchart for TanDEM–X processing.

Finally, we converted the geocoded, unwrapped phase ($\varphi$) to height values ($H'$) as follows:

$$H' = \varphi HoA/2\pi \tag{3}$$

These height values represent height change over time, $\Delta H$, but also contain systematic errors in the form of considerable and large-scale bias and tilt caused by software-specific

issues and depend on orbital parameters and phase height ambiguities. The height change can be calculated by removing the systematic errors as follows:

$$\Delta H = H\prime - k_0 - k_1 E - k_2 N, \tag{4}$$

where $k_0$, $k_1$ and $k_2$ are correction parameters for bias and tilting, and $E$ and $N$ are geographical coordinates for easting and northing. To estimate these correction parameters, we manually placed 52 Ground Control Points (GCPs) around the study area. We placed them in no-forest locations which apparently had stable elevation and high coherence, i.e., mainly farmlands, fields and urban no-building sites, by visual interpretation of imagery on Google Satellite Hybrid. We ensured that there was no height change over time at the location of the GCPs by checking that the unwrapped phase was stable and the coherence was high around the points. We spread them fairly evenly over the study area while in particular placing them in overlapping areas between the acquisitions to ensure a seamless mosaic. For each of the three TanDEM-X data sets, we estimated correction parameters with linear regression analyses using the GCPs, where height change over time should be zero, as follows:

$$H'_{GCP} = k_0 + k_1 E + k_2 N + e \tag{5}$$

where $H'_{GCP}$ is the phase height values for the GCPs, and $\varepsilon$ is the residual error. The three regression models had RMSE values between 0.46 and 0.87 m, with a maximum residual value of 1.9 m. We inserted the estimated correction parameters into Equation (4) and obtained final height change values.

We generated a radar layover and shadow mask and inserted missing values for pixels in such areas. In addition, we discarded all extreme $\Delta$H values, i.e., those lower than $-30$ and higher than 30 m. We also mosaicked the coherence magnitude to be used for placing the GCPs. We combined the $\Delta$H from the three processed TanDEM-X data sets into a mosaic of $\Delta$H. This covered the study area, except for a small area in the lower-left corner. In the areas where the three data sets overlapped, we used the arithmetic mean.

*2.5. Height-to-Biomass Conversion*

The estimation of AGB change contained two steps. First, we derived a conversion factor from InSAR height above ground (IH) and AGB, and, secondly, we used this to directly convert the change in InSAR elevation to estimated change in AGB. The notable advantage of this approach is that terrain elevation is required for the sample of plots only and not wall to wall. The first step required the height above ground of the TanDEM-X phase center for the reference plots, which we calculated as the difference between the TanDEM-X DSM and the terrain elevation. For the NAFORMA plots, we had terrain elevation data both based on airborne laser scanning and from dGPS acquired during field work [33]. We obtained two data sets, i.e., one for 2012 and one for 2019. Phase height of GEDI footprints was calculated in the same way.

An important aspect of the NAFORMA and GEDI plots is their small size for using the TanDEM-X DSM. The plot diameters were 30 and 25 m, corresponding to an area of 707 and 491 m$^2$, respectively. The main problem here is the spatial mismatch between the reference data and the TanDEM-X data. For NAFORMA plots, the entire biomass of a tree is assigned either inside or outside the plot based on the position of the center of the trunk, while the phase center height of TanDEM-X depends on the vertical distribution of scatterers, i.e., mainly the leaves and branches of the tree, which can cover a large area both inside and outside the plot [34]. Both reference data sets have a circular shape, while TanDEM-X consists of raster cells that do not exactly fit the circular outlines. In addition, in a heterogeneous forest, there can be local radar layover and shadowing effects. Additionally, GEDI data have up to ~10 m geolocation uncertainty, which makes colocation with other data sources challenging [35].

These sources of errors generate noise, or random errors. Such random errors should not influence the statistical relationship between AGB and IH. However, for the present

study, the random errors would lead to noisy scatterplots and difficulties in determining whether the relationship is straight and linear or not. Hence, we reduced the random errors by aggregating plot data, and we obtained mean values for AGB and for IH for groups of plots. The NAFORMA plots belonged to clusters, and, hence, using the cluster means in the analysis was intuitive anyway.

For GEDI data, we performed clustering that at the same time solved another issue. The GEDI data were unevenly distributed over the area, where some parts of the study area had no GEDI footprints while other were densely littered with footprints. To make a more balanced representation of different parts of the study area, and to reduce random errors of single footprints, we split the area up into quadratic grid cells and made aggregated means of AGB and IH for each grid cell. We tested a range of gridding from $2 \times 2$ to $32 \times 32$ and selected $8 \times 8$ as the most suitable one.

A ratio between two random variables does not have a defined standard deviation. To cover the need for uncertainty estimates, we estimated the standard error of the ratios, i.e., conversion factors, in two different ways. For the NAFORMA plots, we had two ratio estimates from the two years, and we estimated the SE based on these two estimates. For the GEDI plots, we used a bootstrapping approach. We generated 100 data sets by random sampling with replacement from the grid cells. From each data set, we calculated the ratio between mean AGB and mean IH, and we used the standard deviation between these ratios as an estimate of the standard error of the ratio, or conversion factor.

We described the relationship between AGB and IH using the ratio between their mean values. In this way, the conversion from IH to AGB is based on this ratio, or we can call it proportionality value. This approach has the advantage that it ensures a straight line through the origin, and, in addition, it is an unbiased estimator for AGB.

## 2.6. Geography of Change

We visually assessed the geography of change by focusing on the spatial distribution of pixels and clusters of pixels with a large decrease in elevation. This should represent areas that had forest harvesting, in particular clear cuts, carried out during the eight years. We assessed this based on the presence of forests and the distance from Liwale town, partly based on coherence magnitude and partly on Sentinel-2 RGB imagery. We used a Sentinel-2 data set acquired on 19 June 2020 with a cloud cover of 8%.

## 2.7. Estimating Above-Ground Biomass Change

We used the mean value of the AGB change from NAFORMA plots as a target value and compared this with the predicted AGB change based on InSAR elevation change. We supplemented this with scatterplots showing AGB and InSAR elevation change for single plots and cluster means to gain an understanding of the accuracy of the method for small areas. We used a leave-one-out approach for estimating accuracy. In addition, we also estimated the AGB change wall to wall, including uncertainty estimates. In this latter case, we did not know the true value, and the idea behind this was to demonstrate the application of the method for an area wall to wall.

For analyses of the change in AGB for NAFORMA plots, we included an analysis where we excluded the largest residuals. The rationale behind this was the difference in timing between field inventory and TanDEM-X acquisitions. The study area was not only forest, but contained urban, residential and farmland areas around Liwale town. We can assume that there is disturbance of the forest going on constantly, for firewood gathering and logging to increase the farmland. In the beginning of the eight-year study period, the field plot inventory was carried out mainly in February, while the TanDEM-X acquisitions were centered in May. During the three-month difference period, the disturbance taking place after the field inventory and before the TanDEM-X acquisitions will represent a mismatch between the data sets. This is particularly the case for clear cuts, which would have large AGB values from the field inventory, while the TanDEM-X elevation would correspond to no biomass. At the other end of the period, there was about seven months

of temporal mismatch. Together, the two mismatching periods made up 10 months for a period of 8 years, or 10% of the time. It is possible that the largest 10% of residuals can be attributed to this temporal mismatch. Hence, we discarded the largest 10% of residuals in terms of absolute value and carried out supplementary analysis on these plots only.

## 3. Results

### 3.1. Height-to-Biomass Conversion

The proportionality factors between AGB and InSAR height on NAFORMA plots were 13.8 and 14.2 in 2012 and 2020, respectively (Figure 3). The mean value of this was 14.0 t/ha/m. In comparison, ordinary least-squares regression models resulted in similar values, i.e., regression slopes of 14.4 and 14.5, respectively, and minor intercepts of 1.62 and −1.46. The $R^2$ values were 0.49 and 0.54, respectively, for 2012 and 2020, and the residual errors (RMSE) were 43.1 and 41.5 t/ha in AGB. On average, this corresponded to an RMSE of 84% relative to the mean AGB stock of 49.5 t/ha calculated for both years together. The RMSE values were strongly influenced by a few high-biomass plots. By removing plots above 300 t/ha, the RMSE was reduced to 27 t/ha. One high-influence point in 2020 is highlighted in red in the upper-right corner of the scatterplot (Figure 3b). This plot contained one large *Brachystegia bussei* tree with a diameter at breast height of 120 cm and an estimated height of 27 m. We do not know the location of this tree inside the plot, but we can imagine that the main scattering elements in this tree's canopy for the TanDEM-X microwaves, coming at an incidence angle of 34°, may well be outside of the plot. In addition, effects from local layover, shadowing and spatial smoothing during the processing for such a large, solitary tree may well produce errors. Based on the GEDI L4A data, the proportionality factor was 11.4, based on all footprints and the mean values of AGB and IH (Figure 3c).

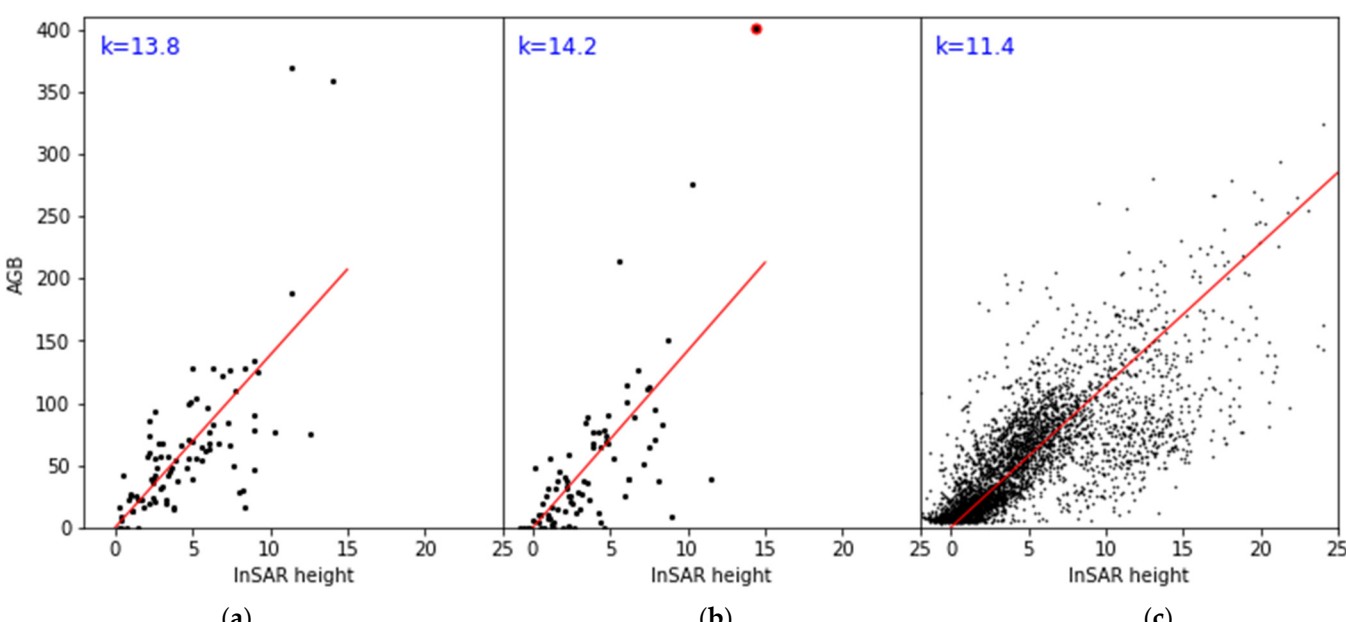

**Figure 3.** Relationship between above-ground biomass (AGB, t/ha) and InSAR height (m) for (**a**) NAFORMA 2012 plot data and TanDEM-X global DEM, (**b**) NAFORMA 2020 plot data and TanDEM-X 2019 where one high-influence point is highlighted in red and (**c**) GEDI L4A 2019 and 2020 footprints and TanDEM-X 2019. Black dots are plots and red line represents proportionality.

After clustering the plots, we obtained relationships that complied to the prerequisites of the present approach by being straight and linear, passing through origin and being similar at both points in time (Figure 4). The heteroscedasticity seen in Figure 3 was removed by the clustering, indicating that random errors caused by small plots were the

main reason for the heteroscedasticity. The two obtained conversion factors, or slopes, were moderately different. They differed by 15%. The NAFORMA-plot-based conversion factor was higher (14.0) than the GEDI-based factor (12.1). The plot-based factor had an estimated standard error that was small, being only 0.18. It was considerably smaller than the one based on GEDI.

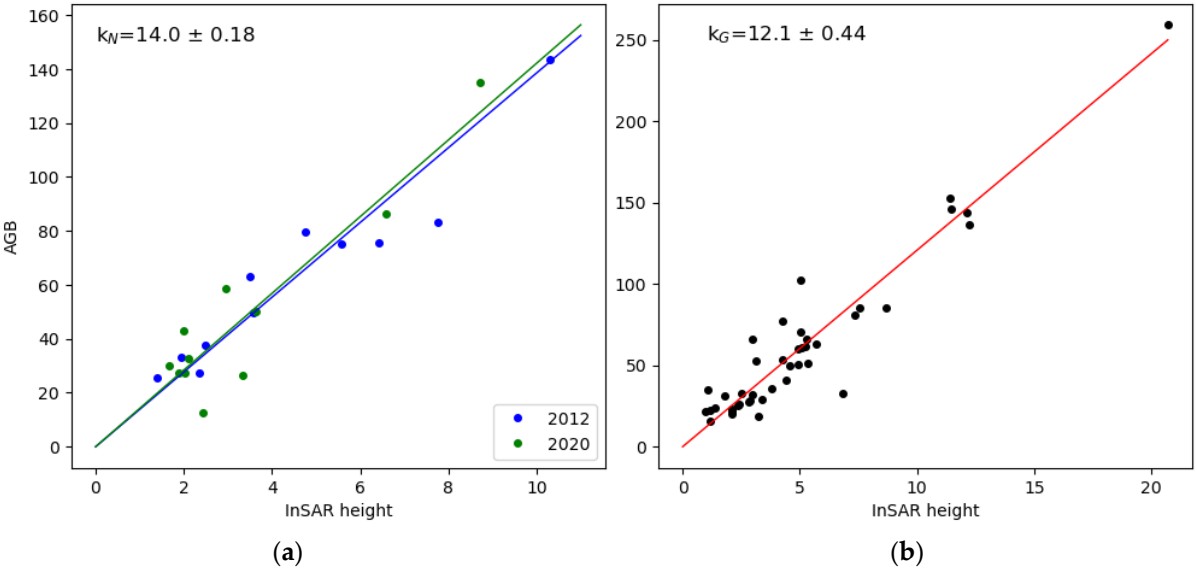

**(a)**                                                     **(b)**

**Figure 4.** Relationship between above-ground biomass (AGB, t/ha) and InSAR height (m) after clustering for (**a**) NAFORMA plots and (**b**) GEDI. The lines represent the proportionality, and are the ones we used as the final conversion factors, $k_N$ for NAFORMA plots and $k_G$ for GEDI L4A footprints.Blue color on lines and dots represent 2012, green represents 2020, and black dots with red line represent GEDI footprints.

The clustering of GEDI footprints into $8 \times 8$ cells worked well. Out of the 64 $8 \times 8$ grid cells, 42 had at least one footprint. This means that we had a fairly large number of clusters and, at the same time, a fairly good coverage of the area (66%, Table 2). When we split the area up in other grids, the results were similar. For the alternatives with small cells, the advantage was a high number of obtained conversion values, while, on the other hand, the disadvantage was that a small fraction of the study area was covered. In addition, we obtained several extreme values for small cells containing only one or two GEDI footprints.

**Table 2.** Results for alternative clustering settings for GEDI data, where $N$ = total number of cells, % GEDI area means the percentage of cells having at least one GEDI footprint and $k$ is the conversion factor.

| Alt. | $N$ | % GEDI Area | $k$ |
|---|---|---|---|
| $2 \times 2$ | 4 | 100 | 11.5 |
| $4 \times 4$ | 16 | 88 | 11.9 |
| $8 \times 8$ | 64 | 66 | 12.1 |
| $16 \times 16$ | 256 | 39 | 11.6 |
| $32 \times 32$ | 1024 | 26 | 11.7 |

The next processing step for GEDI data, the bootstrapping, also worked well. This was demonstrated firstly by the identical estimate for the conversion factor (12.1) provided. We also confirmed the correctness of the bootstrap approach using ordinary linear regression. The regression slope based on the $8 \times 8$ cell means was $11.8 \pm 0.60$, while separate regression models for each of the 100 data sets produced 100 regression slopes with a mean value of 11.6 and a standard deviation of 0.59, while the mean SE of the slopes was 0.62.

### 3.2. Geography of Change

The 2012–2020 estimated AGB change contained a clear spatial pattern dominated by areas that experienced a decrease. These areas that experienced a decrease appeared as an outline of logged areas with Liwale town in the center (Figure 5). This suggested an expansion of urban and farmland areas into the remaining forest areas around the town. The areas that experienced a decrease corresponded fairly well to the apparent no-vegetation areas with a red-brown, light color in the Sentinel-2 RGB imagery. We should not expect such a complete correspondence because some clear cuts from the beginning of the 8-year period might have regrown with new vegetation. The remaining forest appeared as dark green in the Sentinel-2 RGB imagery. Some patches appeared to have experienced forest growth with increased AGB. These changes cannot be verified by the Sentinel-2 RGB image.

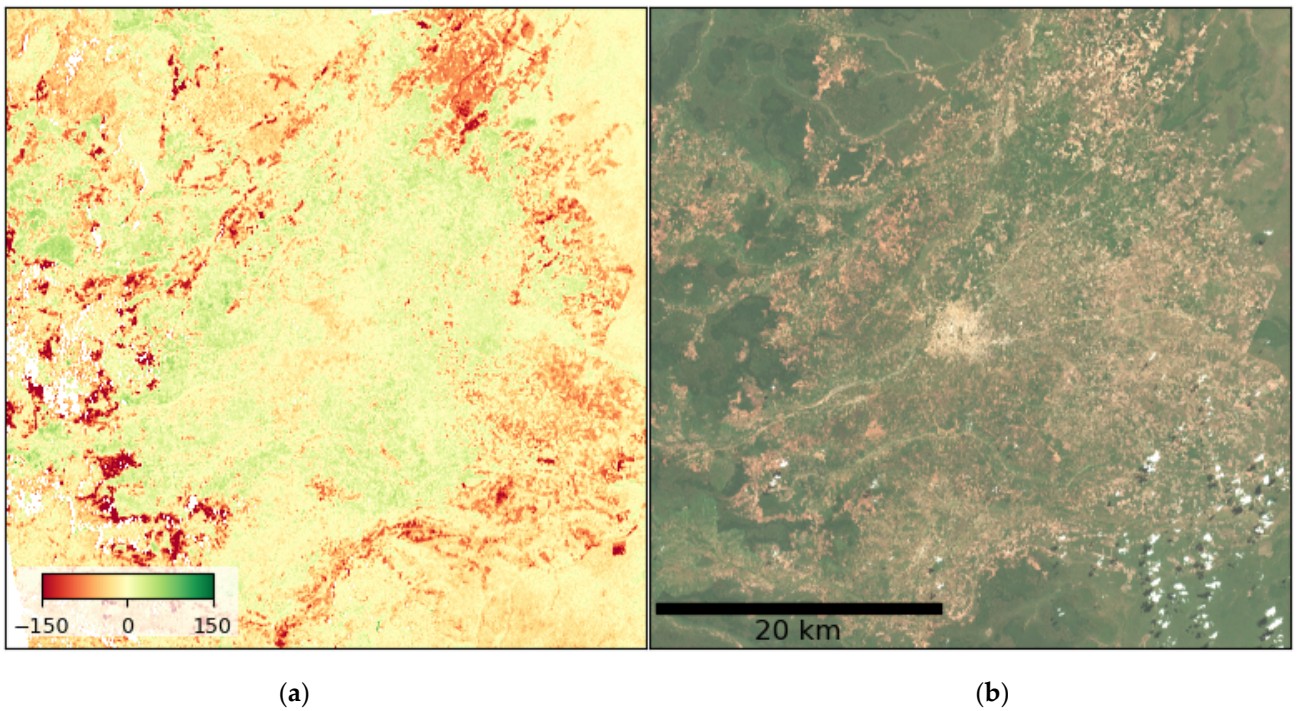

(**a**)  (**b**)

**Figure 5.** (**a**) Geography of estimated AGB change 2012–2020 [t/ha], (**b**) Sentinel-2 RGB imagery acquired on 19 June 2020.

### 3.3. Estimated Above-Ground Biomass Change

Temporal change in AGB could be estimated within 5–10% error margins. The mean change for the NAFORMA plots was −14.5 t/ha, while the predicted mean change was −16.0 when using the plot-based conversion factors and −13.8 with the one based on GEDI (Table 3). This is based on the mean InSAR elevation change of −1.14 m which we converted to AGB change. This means that the true mean change of the NAFORMA plots was well within the range given as the predicted mean value and the standard error of the mean and close to the true mean. In this case, the standard error was based on a leave-one-plot-out approach. We repeated this calculation excluding the largest 10% of residuals, and the results were slightly improved. The mean AGB change was now −12.6 t/ha, and the estimate based on the NAFORMA plot conversion factor was −13.2 ± 0.17, while, for GEDI, the corresponding estimate was −11.4 ± 0.42. This means that the results were fairly robust, having small influence from extreme points.

For the study area, we had no ground truth. However, the mean InSAR elevation change was almost identical to that of the NAFORMA plots. Hence, we can assume that the accuracy for the entire area would be the same.

**Table 3.** Change in TanDEM-X elevation, $\Delta DSM_{TDX}$ [m], and change in above-ground biomass, $\Delta AGB$ [t/ha], for ground truth and estimates $\pm$ SE. Note that we do not have a ground truth for the entire area. The two conversion factors $k_N$ and $k_G$ are estimated from NAFORMA plot data and GEDI L4A footprint data, respectively.

|  | NAFORMA Plots | Study Area |
|---|---|---|
| $\Delta DSM_{TDX}$ | $-1.14$ | $-1.10$ |
| $\Delta AGB$, ground truth | $-14.5$ | |
| $\Delta AGB = k_N \ \Delta DSM_{TDX}$ | $-16.0 \pm 0.21$ | $-15.4 \pm 0.20$ |
| $\Delta AGB = k_G \ \Delta DSM_{TDX}$ | $-13.8 \pm 0.50$ | $-13.3 \pm 0.48$ |

A closer look at the temporal change for individual NAFORMA plots revealed some clear outliers (Figure 6). This was as expected due to a non-perfect time correspondence between the field inventory and satellite data acquisition. After excluding the largest 10% of residuals due to a possible mismatch in timing of the satellite acquisitions and the field inventory, there was a Pearson correlation coefficient between the AGB change and the InSAR elevation change of 0.70 (Figure 6). There was a moderate residual scattering around the conversion factor line, which is attributable to the random errors caused by small plots. The RMSE was 20.9 t/ha when we excluded the plots with the worst 10% of residuals. With the leave-one-out cross validation, we excluded the NAFORMA plots one by one and, in each case, calculated a conversion factor between InSAR height and AGB separately for 2012 and 2020, averaged them to a conversion factor value and calculated the RMSE. The result of this was a mean conversion factor of 13.1 and an RMSE of 20.3 t/ha.

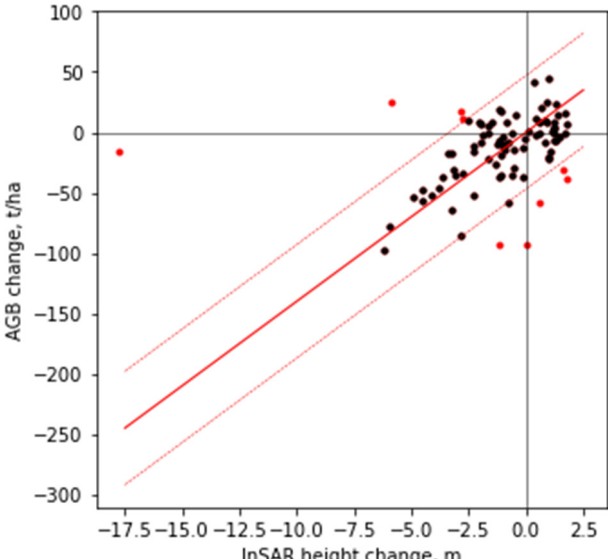

**Figure 6.** Relationship between temporal change in AGB and InSAR height from 2012 to 2020 for the NAFORMA plots. The dots represent plots where the largest 10% of residuals are identified in red and between red hatched lines. The red solid lines represent the NAFORMA plot conversion factor $k_N$.

## 4. Discussion

### 4.1. Geography

The geography of changes appeared to be right, based on visual correspondence with the Sentinel-2 RGB and based on Liwale town located in the center of an apparent expansion of deforested area. Generating a correct geographical representation of real forest changes including detection of areas experiencing both growth and disturbance appears to be easily achievable with this method, as already shown in former studies [18,20].

*4.2. Conversion from InSAR Height to Above-Ground Biomass*

In this study, we used a simple conversion from InSAR phase height to AGB, i.e., using a fixed proportionality factor. This approach has been compared to a more sophisticated method in a boreal forest where both forest height and density were estimated and combined [36]. In general, that latter approach performed better in terms of larger $R^2$ values. However, the difference was variable, and, in one case, it was the opposite. In any case, an approach using a conversion factor is the only approach that can be employed for repeated TanDEM-X elevation data in areas without wall-to-wall Digital Terrain Models (DTMs) available, and, in the present study, it was clearly demonstrated that the simple linear model represented a good fit to the data.

The conversion factors we obtained between InSAR height and AGB are reliable because similar values have been found in other studies across forest types. In the present study, we found the two values 12.1 and 14.0 t/ha/m, which are close to those of previous studies, varying from 11.9 [14] to 14.1 [12], from the same miombo woodland in Tanzania. It was somewhat larger, 18.4 t/ha/m, in a very-high-AGB-density forest in northeast Tanzania [13], and it was 13.0 based on data from a tropical forest in Brazil [37] and 14.9 in a Norway spruce forest in Norway [20]. The estimation of this conversion factor can be performed in different ways, and different methods yield slightly different values. If we look closer at the GEDI-based conversion factor in this study, it was 12.1 t/ha/m based on proportionality, 11.8 with ordinary least-squares regression and 12.0 with a no-intercept regression model. In most earlier studies, the conversion factor was estimated by no-intercept regression models. Although different methods may yield fairly similar conversion factors, the regression-based models have problems. An ordinary regression model has an intercept, which will lead to non-zero, or ambiguous, AGB estimates in no-forest areas. A no-intercept regression model has the weakness that it does not represent an unbiased estimator of AGB. Hence, using the proportionality as we did in this paper is an improvement. For the present approach, where we derive the change in AGB from the change in surface model height, it is required that the relationship between AGB and InSAR height is linear and passing through the origin. If those two criteria are not met, we would need to know the terrain elevation wall to wall of the study area.

This GEDI-based conversion factor was somewhat lower than the ones based on the NAFORMA plots. The difference may be attributable to different coverage and uneven sample intensity over the area, as well as positional uncertainty on the GEDI footprints.

*4.3. Comparability of the Two Points in Time*

The present approach requires that the penetration of the SAR microwaves down into the forest canopy is similar at both points in time. The basis for this is that the relationship between InSAR height and biomass is indirect because the short-wavelength SAR mainly interacts with small objects like leaves and small branches. Normally, the number of such small objects would be strongly related to the AGB, which is mainly made up of the trunks. However, a variation between leaf-on and leaf-off conditions affects the relationship [38]. In this study, the relationship was almost identical at the two points in time, and, apparently, the miombo woodland was in a leaf-on stage around the time of the acquisitions both in 2012 and 2019. However, this issue may limit the large-scale application of this method based on the two TanDEM-X global DEMs. The former TanDEM-X global DEM was made by combining data from many acquisitions, and each pixel does not have any date of acquisition tag. The acquisition dates are given in the metadata of the DEM tiles, which may help, but the dates are not spatially explicitly given. First, this issue would be present for forests with seasonal leaf-on and leaf-off variations. We can imagine that this would be the case for savannah forests and for temperate broadleaf forests. Secondly, this issue would also occur in boreal forests, where the phase height is strongly affected by frost versus no-frost, and, in general, dryness can affect the microwave extinction through the canopy [22,39,40]. However, in tropical rainforest, the phase height appears to be stable across weather types and seasons [21] so the method appears promising for such forest.

*4.4. Vertical Adjustment*

The estimation of AGB change relies on one subjective processing step, i.e., the vertical bias adjustment. In this study, we processed the three TanDEM-X single-look complex data sets against the former global DEM. In principle, the interferogram should represent elevation change only. However, as seen in other studies, there were errors representing both vertical bias and tilting of the DEM. We placed GCPs around the study area, subjectively in areas that appeared to be stable, no-forest locations based on optical satellite data, and used them to correct both the tilting and bias in the three TanDEM-X data sets. This subjective processing step could prevent a large-scale, fully automatic application of the method. However, it appears that the tilting and bias errors are partly attributable to software-specific processing methods. It is also possible that this adjustment could be performed automatically. More important, however, is that, with the release of the new global DEM based on TanDEM-X, we can assume that these errors are removed.

*4.5. GEDI and TanDEM-X Combination*

This study suggests that the GEDI L4A product can replace AGB from field inventory plots for model training and validation. The conversion factors based on NAFORMA plots and GEDI footprints were fairly similar. However, the present study is a special case here. In general, it is uncertain whether we would obtain equally high similarity in other areas. The GEDI AGB estimator is a generic model relating field AGB to simulated GEDI from airborne LiDAR based on a global compilation of reference data sets [31]. For the present study area of Tanzania, the AGB estimator is trained using the NAFORMA plot data from 2012. There is a lack of other available airborne laser scanning data from East Africa; indeed, the broader NAFORMA area is the only training data for African deciduous forests, and, therefore, the field-AGB-to-GEDI model is more locally tuned to this site than any other model in the GEDI product [31]. Tanzania's NAFORMA data set is somewhat unique for tropical countries. We recommend a further study on the conversion factor between phase height and AGB for forest types within the near-global coverage of GEDI. By combining the huge GEDI data set and global TanDEM-X data acquired close in time, it should be possible to study the variation of the AGB–InSAR height relationship. On the other hand, the conversion factor appears to be similar for a wide range of forest types, and this reduces the importance of having a specific conversion factor for the forest type in question.

**5. Conclusions**

In conclusion, temporal change in miombo woodland biomass is closely related to change in X-band, and InSAR elevation, and this enabled both an accurate mapping of the geography of temporal AGB change and a quantification wall to wall within 5–10% error margins. The combination of TanDEM-X and GEDI has the potential for a spatially explicit and near-global estimation of temporal change in AGB. However, care needs to be taken to avoid errors based on effects of leaf-on versus leaf-off canopy conditions and frost versus no-frost.

**Author Contributions:** Conceptualization, S.S.; methodology, S.S., O.M.B., T.G. and E.N.; software, S.S. and O.M.B.; validation, S.S.; formal analysis, S.S. and O.M.B.; investigation, S.S.; resources, S.S., O.M.B., T.G., E.N., L.I.D. and P.B.; data curation, S.S. and O.M.B.; writing—original draft, S.S. and O.M.B.; writing—review and editing, S.S., O.M.B., T.G., E.N. and L.I.D.; visualization, S.S.; supervision, S.S.; project administration, S.S.; funding acquisition, S.S. All authors have read and agreed to the published version of the manuscript.

**Funding:** This research was funded by the European Space Agency (ESA) through the Prodex grant number 4000132157 for the project "Forest INSAR height and biomass change—FORINSAR".

**Data Availability Statement:** The data presented in this study are openly available in "Data for Remote Sensing: Biomass Change Estimated by TanDEM-X Interferometry and GEDI in a Tanzanian Forest" at doi.org/10.18710/UZOUB0.

**Acknowledgments:** We thank the German Aerospace Center (DLR) for providing the TanDEM-X data through the two projects sossos_XTI_LAND0333 and sossos_XTI_VEGE0315. We thank Eliakimu Zahabu, Sokoine University of Agriculture, Tanzania, for organizing the 2020 field inventory.

**Conflicts of Interest:** The authors declare no conflicts of interest.

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
