# Peer review of "Biomass Change Estimated by TanDEM-X Interferometry and GEDI in a Tanzanian Forest"

_remotesensing, doi:10.3390/rs16050861_

Round 1

Reviewer 1 Report

Comments and Suggestions for Authors

This paper is based on TanDEM-X, NAFORMA, and GEDI data. By establishing a regression relationship between tree height and above-ground biomass, it achieves the monitoring of biomass changes. The GEDI footprint data is integrated into wall-to-wall biomass retrieval products through a spatial aggregation scale conversion method for detection and application, which is innovative, and the conclusions are reliable. However, some key methods are described vaguely and require further modification. Specific recommendations are as follows:

Please add necessary legends and scale bars to Figures 1 (a) and (b) to help readers clearly understand the size of the study area.

In section 2.2, has the GEDI L4A data selected in this paper undergone necessary quality screening? What indicators were used for screening? How many footprints were used specifically? Additionally, the selected GEDI data distinguishes four time windows, so please differentiate the GEDI data for different time windows in Figure 1 (b) using different colors.

Section 2.3 provides a very detailed description of the calculations, but it seems more reasonable to place the calculation part in a separate subsection. Furthermore, there are many steps in the data processing and calculation workflow of this paper. Please supplement a flowchart to present the overall calculation process involved in this paper.

In section 2.5, the description of how to eliminate errors caused by differences in plot sizes and the uncertainty of spatial coordinates is very critical in the research methodology. However, there are two issues that need further clarification. The first issue is why the study area is divided into 8×8 grids; is there a rationale for this? Some discussion can be found in section 3.2 later, but it is too brief. Please provide a more detailed description in this section. The second issue is whether a schematic diagram can be created for this section to help readers better understand the efforts made in this section.

Please add necessary legends and scale bars to Figures 2 (a) and (b) as well.

In section 3.2, line 353, it is recommended to supplement specific data for different partitioning methods such as n 4×4, 16×16, and 64×64. This can be presented in the form of figures or tables.

Comments on the Quality of English Language

Moderate editing of English language required.

Reviewer 2 Report

Comments and Suggestions for Authors

This manuscript explored the potential of TanDEM-X for forest biomass change estimation and the potential of GEDI as an alternative to filed inventory for this purpose, and then contributed to global forest biomass change using TanDEM-X and GEDI. As a complement, the main difference between this work and previous work is the new method to obtain the conversion from height to biomass using a ratio rather than a linear regression (line 80 & lines 271-274), which makes this work less innovative. More importantly, the main problem is as follows.

It seems that an important part of the result validation is missing. The validation consists of two parts, the validation of biomass change estimation and height change. For the biomass change estimation (Table 2), the mean values alone are not convincing. In addition, the comparison between them is problematic as K was estimated from the ground truth and TDX data. In this case, cross validation is suggested. It is also suggested to add the validation between the height change extracted from the InSAR and the reference data.

In section 2.3. it is weird to construct the relationship between dH and coordinates directly (Equation 4). In this part, the height change was calculated, but in the result part, the height was used, which is inconsistent if no more information is provided.

It will be better to provide the biomass change map for the entire study areas. In addition, figure 5 can be merged into one figure.

Line 254-255: The cluster method should be elaborated as it affects the results of height to biomass estimation.

Figure 2, a legend is necessary.

Comments on the Quality of English Language

Moderate editing of English language required, such as the incomplete sentence in line 220.

Round 2

Reviewer 1 Report

Comments and Suggestions for Authors

This artical is acceptable for me now.

Comments on the Quality of English Language

English language quality is acceptable

Reviewer 2 Report

Comments and Suggestions for Authors

The author has successfully addressed the issues raised in the previous round, and I recommend its publication.